# Switching molecular recognition selectivities by temperature in a diffusion-regulatory porous material

Yan Su[1], Ken-ichi Otake [2], Jia-Jia Zheng[3], Hong Xu [4], Qing Wang[5], Haiming Liu [5], Fei Huang[6], Ping Wang[7], Susumu Kitagawa [2] & Cheng Gu [1,7]

Over the long history of evolution, nature has developed a variety of biological systems with switchable recognition functions, such as the ion transmissibility of biological membranes, which can switch their ion selectivities in response to diverse stimuli. However, developing a method in an artificial host-guest system for switchable recognition of specific guests upon the change of external stimuli is a fundamental challenge in chemistry because the order in the host-guest affinity of a given system hardly varies along with environmental conditions. Herein, we report temperature-responsive recognition of two similar gaseous guests, $CO_2$ and $C_2H_2$, with selectivities switched by temperature change by a diffusion-regulatory mechanism, which is realized by a dynamic porous crystal featuring ultrasmall pore apertures with flip-flop locally-motive organic moiety. The dynamic local motion regulates the diffusion process of $CO_2$ and $C_2H_2$ and amplifies their rate differences, allowing the crystal to selectively adsorb $CO_2$ at low temperatures and $C_2H_2$ at high temperatures with separation factors of 498 ($CO_2/C_2H_2$) and 181 ($C_2H_2/CO_2$), respectively.

Molecular recognition plays a vital role in supramolecular chemistry[1,2], in which specific affinity among molecules allows for the construction of high-order assemblies and stimuli response[3]. Usually, recognition of a sole guest with the largest affinity from the multiple-guest mixture can be achieved[4], whereas specific recognition of different guests under varied environmental conditions remains challenging. A basic scientific issue is a limitation from thermodynamics, where the order of host-guest affinities hardly changes with environmental conditions in a given host-multiple guest system[2]. On the other hand, specific recognition of different guests is highly desired, and such "smart" host materials can be widely applied in various fields such as molecular machines[5], sensors[6], gas separation[7], and drug delivery[8]. To achieve

specific recognition switchable to different guests, chemists attempted to change the host-guest affinity using stimuli-responsive guests[9,10], whose chemical structures or molecular conformations change with external stimuli. However, such a strategy is only limited to cyclodextrin-azobenzene[11], cyclodextrin-benzimidazole[12], and cyclodextrin-ferrocene systems[13], whose host-guest affinities can be switched by light, pH, and redox, respectively. A simple and effective strategy for switchable molecular recognition that breaks through the limitation of thermodynamics has not been proposed so far.

Porous coordination polymers (PCPs)[14–17] or metal-organic frameworks are highly designable material platforms whose structural softness and pore environment can be tailored for molecular

[1]State Key Laboratory of Luminescent Materials and Devices, Institute of Polymer Optoelectronic Materials and Devices, South China University of Technology, Guangzhou 510640, P. R. China. [2]Institute for Integrated Cell-Material Sciences, Kyoto University, Kyoto 606-8501, Japan. [3]Laboratory of Theoretical and Computational Nanoscience, National Center for Nanoscience and Technology, Chinese Academy of Sciences, Beijing 100190, P. R. China. [4]Institute of Nuclear and New Energy Technology, Tsinghua University, Beijing 100084, P. R. China. [5]School of Physical Science and Technology, ShanghaiTech University, Shanghai 201210, P. R. China. [6]ReadCrystal Biotech Co., Ltd., Suzhou 215505, P. R. China. [7]College of Polymer Science and Engineering, State Key Laboratory of Polymer Materials Engineering, Sichuan University, Chengdu 610065, P. R. China. ✉e-mail: kitagawa@icems.kyoto-u.ac.jp; gucheng@scu.edu.cn

recognition. Despite the recent progress, most of the molecular recognition in PCPs is based on thermodynamic adsorption, which is inaccessible for switchable recognition. On the other hand, the control over guest-transport kinetics allows precise discrimination of similar guests[18,19], yet the strategy for switchable molecular recognition is still not proposed. Herein, we present switchable molecular recognition of two similar gaseous guests, $CO_2$ and $C_2H_2$, only by temperature, without changing their host-guest affinity. This is achieved by regulating the gas diffusion and amplifying their rate differences using a locally dynamic PCP, in which flip-flop molecular motions of the ligand provide gate functionality. $CO_2$ exhibits faster diffusion than $C_2H_2$, rendering the preferential adsorption of $CO_2$ and $CO_2/C_2H_2$ selectivity higher than 1 at the thermodynamic non-equilibrium state at low temperatures. By contrast, $C_2H_2$ possesses higher adsorption affinity than $CO_2$, resulting in selective adsorption of $C_2H_2$ and $C_2H_2/CO_2$ selectivity higher than 1 at the thermodynamic equilibrium state at high temperatures. Therefore, significant temperature-dependent adsorption behaviors are observed for $CO_2$ and $C_2H_2$, with both striking $CO_2/C_2H_2$ and $C_2H_2/CO_2$ selectivities at low and high temperatures, respectively (Fig. 1a).

## Results

### PCP synthesis and structural analyses

We designed a bee-type ligand comprising [1,1':3',1''-terphenyl]-3,3''-dicarboxylic acid and phenothiazine-5,5-dioxide (OPTz) moieties (OPTz-t3da), with the later moiety exhibiting effective waggling motion; the OPTZ moiety can waggle around its equilibrium position by ~20° with small energy increases by <25 kJ mol$^{-1}$ (Supplementary Fig. 1). Such a waggling motion of ligand leads to the dynamical opening and blocking of channels in PCP crystal, which is thus termed flip-flop dynamic crystal (FDC). The as-synthesized PCP, termed **FDC−3** (Supplementary Figs. 2–4, Supplementary Table 1), was subjected to solvent exchange and thermal activation to afford its activated phase (**FDC−3a**, Supplementary Figs. 5–10, Supplementary Table 2). The crystal structure of **FDC−3a** was determined by the continuous rotation electron diffraction (cRED) technique (Supplementary Fig. 5). Activation caused a structural transformation of **FDC−3** into a two-fold interpenetrated, 3, 6-connected rutile (rtl) topological framework with small yet compact pores (Fig. 1, b, c, Supplementary Fig. 6). The pore aperture was surrounded by one OPTz moiety and one O atom on carboxylic acid to form an ultrasmall gate of 2.9 Å in size, which was expected to be gradually enlarged by the thermal flipping of OPTz moiety, allowing the diffusion of gases at high temperature and blocking them at low temperatures.

### Gas sorption

**FDC−3a** adsorbed $CO_2$ and $C_2H_2$ and showed negligible adsorption for other gases, including $N_2$, $CO$, $O_2$, $Ar$, $C_2H_4$, and $C_2H_6$, in a wide temperature range (Fig. 2a, Supplementary Fig. 11). The adsorption amounts for both $CO_2$ and $C_2H_2$ substantially increased as increasing the temperature, as shown in their adsorption isotherm curves (Supplementary Fig. 12). Taking $CO_2$ as an example (Supplementary Figs. 12–14), the adsorption amount increased from 25 to 41 mL g$^{-1}$ as the temperature was increased from 200 to 240 K and then decreased to 5 mL g$^{-1}$ as the temperature was further increased to 370 K. Therefore, the temperature of maximum adsorption amounts ($T_{max}$) of $CO_2$ was 240 K, and similarly, the $T_{max}$ of $C_2H_2$ appeared at 320 K; the $T_{max}$ values for both $CO_2$ and $C_2H_2$ were substantially higher than their boiling-point temperatures ($T_{bp}$). This is a distinctive adsorption feature in which the initial adsorption was promoted by temperature, making a sharp contrast to common gas adsorption under thermodynamic equilibrium, in which the adsorption amount monotonously decreases as increasing the temperature. Additionally, obvious desorption hysteresis was observed for $CO_2$ (200–300 K) and $C_2H_2$ (200 to 360 K) in their sorption isotherms, which was characteristic of the

diffusion-regulatory pore systems in PCPs[18,19]. These results further indicated that the diffusions of $CO_2$ and $C_2H_2$ were regulated in the temperature ranges of 200 to 300 K and 200 to 360 K, respectively, showing that the adsorption of $CO_2$ was controlled by kinetics and thermodynamics at low (200 to 300 K) and high (320 to 360 K) temperatures, respectively, whereas the adsorption of $C_2H_2$ was constantly controlled by kinetics. The temperature-assisted adsorption behavior was controlled by kinetics, in which the diffusion of gases was impeded by low temperature, whereas the diffusion was gradually promoted by raising the temperature. Remarkably, the $T_{max}$ values of $CO_2$ and $C_2H_2$ were largely different by 80 K, although they had exactly the same kinetic diameters and very similar molecular sizes and polarizabilities (Supplementary Table 3). Therefore, the selectivity can be switched by temperature; **FDC−3a** preferably adsorbed $CO_2$ in the 200 to 280 K range, whereas it reversely selected $C_2H_2$ in the 290 to 370 K range. The maximum adsorption ratios for $CO_2/C_2H_2$ and $C_2H_2/CO_2$ were 2.9 (at 220 K) and 3.6 (at 350 K), respectively (Fig. 2a).

Although the above-mentioned sorption curves already revealed an apparent difference in the adsorption amounts of $CO_2$ and $C_2H_2$, they were not able to reflect the differences in the adsorption kinetics. Therefore, we performed kinetic adsorption of $CO_2$ and $C_2H_2$ at different temperatures by **FDC−3a** (Supplementary Fig. 15). The adsorption amounts for both $CO_2$ and $C_2H_2$ were lower than the amounts in their corresponding isobar curves, whereas the $T_{max}$ for both $CO_2$ and $C_2H_2$ slightly shifted to higher temperatures. These results indicated that the kinetic factors were key to affecting the adsorption behaviors of $CO_2$ and $C_2H_2$. On the other hand, the switching of selectivity was also observed in the kinetic adsorption, which further proved that the cooperativity of diffusion regulation and host-guest interaction caused the temperature-switchable selectivity even in the kinetic conditions.

### IAST selectivities and diffusion rates

We employed the ideal adsorbed solution theory (IAST) to predict the selectivity of a $CO_2/C_2H_2$ mixture for practical separation (Fig. 2b, Supplementary Fig. 16, Supplementary Tables 4, 5). The IAST selectivity in **FDC−3a** was temperature-manipulated: in the temperature range of 200 to 280 K, **FDC−3a** was $CO_2$-selective with the $CO_2/C_2H_2$ selectivity higher than 1; in the temperature range of 300 to 360 K, **FDC−3a** turned to $C_2H_2$-selective with the $CO_2/C_2H_2$ selectivity lower than 1 (i.e., $C_2H_2/CO_2 > 1$). Such a temperature-switched selectivity was not observed in other porous materials. Taking an equimolar mixture of $CO_2/C_2H_2$ at 1 bar and various temperatures as an example, the maximum $CO_2/C_2H_2$ selectivity was up to 18.6 at 220 K and the minimal $CO_2/C_2H_2$ selectivity was as low as 0.11 (corresponding to a $C_2H_2/CO_2$ selectivity of 9.5) at 360 K. Even though at a condition of very low feed-gas components (5% $CO_2$ at 200 K and 5% $C_2H_2$ at 360 K), the $CO_2/C_2H_2$ selectivity values remained to be 15.6 (200 K) and 0.2 (360 K), respectively, indicative of the capability of enriching $CO_2$ at low temperatures and $C_2H_2$ at high temperatures. **FDC−3a** exhibited high IAST selectivity for $CO_2$ and $C_2H_2$ at low and high temperatures, respectively, revealing the potential for the selective adsorption of $CO_2$ and $C_2H_2$ in a temperature-controlled manner.

To uncover the essence of the temperature-switched adsorption, we employed the Crank theory to quantify the diffusion rate for every $CO_2$ and $C_2H_2$ adsorption plot from the corresponding isotherms in the 200 to 360 K range, which allowed the production of global T−D$_s$/R$^2$−V and P−D$_s$/R$^2$−V landscapes, where T (K), P (kPa), D$_s$/R$^2$ (s$^{-1}$), and V (mL g$^{-1}$) denote temperature, pressure, diffusion rate, and uptake volume, respectively, where R refers to the radius of an **FDC−3a** particle (Fig. 2, c, d, Supplementary Fig. 17). The landscapes revealed that the diffusion rates for both $CO_2$ and $C_2H_2$ were substantially low at low temperatures, whereas they steadily increased with increasing temperature and pressure, accompanied by the enhanced uptake amounts up to the $T_{max}$ of $CO_2$ and $C_2H_2$. The diffusion rates for $CO_2$ and $C_2H_2$ at 240 K ($T_{max}$ of $CO_2$) and 1 bar were $3.22 \times 10^{-2}$ and $8.29 \times 10^{-3}$ R$^2$ s$^{-1}$,

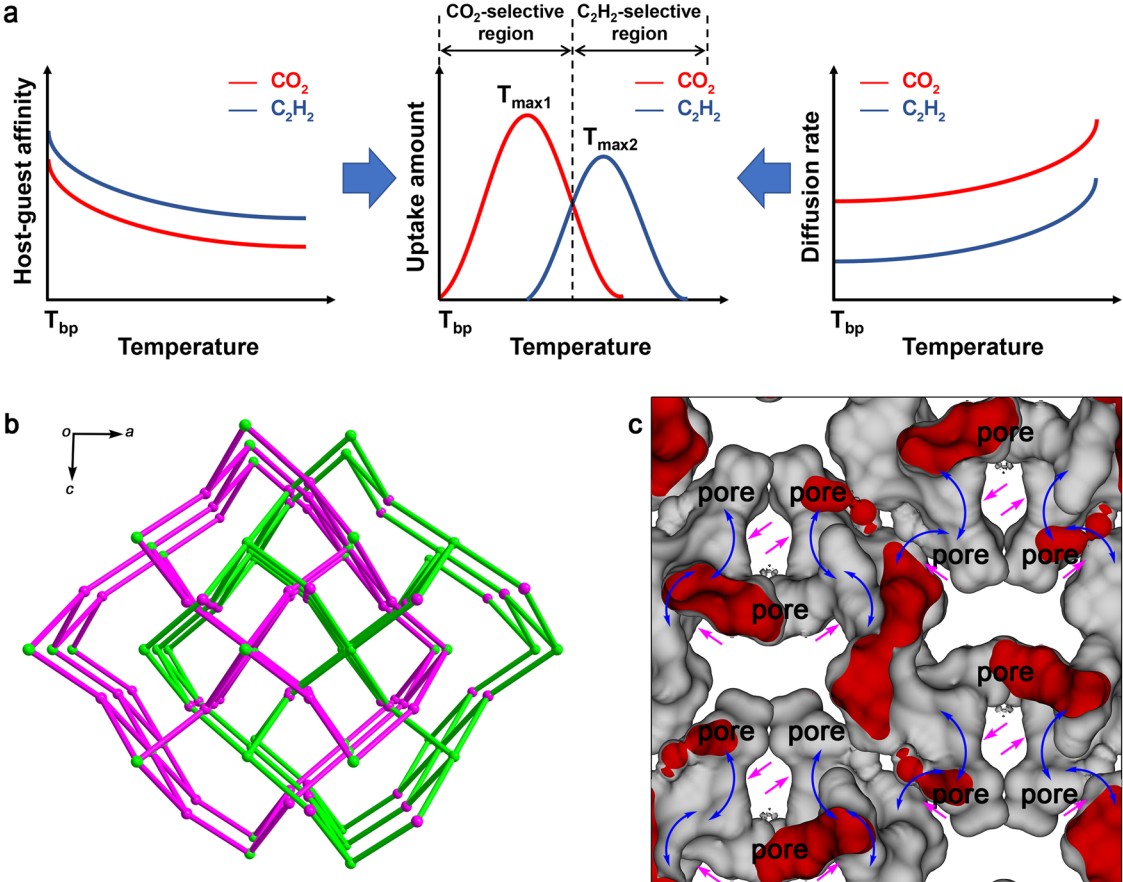

**Fig. 1 | The diffusion-regulatory mechanism for the temperature-switched recognition of $CO_2$ and $C_2H_2$. a** Schematic representation of the mechanism. Left: temperature-dependence of diffusion rates of $CO_2$ and $C_2H_2$. Right: temperature-dependence of host-guest affinities of $CO_2$ and $C_2H_2$. Middle: dynamics manipulation in diffusion-regulatory PCPs (this work); this mechanism involves a pore system featuring diffusion-regulatory functionality that can regulate the diffusion of gases and amplify their rate differences, thereby resulting in a temperature-switched recognition in which the gas with a high diffusion rate but a low affinity is preferentially adsorbed at low temperatures and the gas with a low diffusion rate but a high affinity is selectively adsorbed at higher temperatures. **b** The two-fold interpenetrated, 3, 6-connected rutile (rtl) topology of **FDC−3a**. The $Zn^{2+}$ dual-tetrahedron cluster possesses 6 coordination sites, simplified as a 6-connected node and represented with green balls. The ligand is linked with 3 $Zn^{2+}$ clusters, which is simplified as a 3-connected node and is represented with purple balls. **c** The void in **FDC−3a** visualized by a small probe radius of 0.6 Å. The void volume is 692 Å$^3$ and corresponds to 13.3% of the unit-cell volume. The inner and outer surfaces of the pore are drawn in red and gray, respectively. The pink and blue arrows show the diffusion windows and pathways, respectively.

respectively, whereas the diffusion rates for $CO_2$ and $C_2H_2$ at 320 K ($T_{max}$ of $C_2H_2$) and 1 bar were $7.24 \times 10^{-2}$ and $2.18 \times 10^{-2}$ R$^2$ s$^{-1}$, respectively. The diffusion rates of $CO_2$ were substantially higher than $C_2H_2$ at all temperatures, despite the same kinetic diameters and very similar molecular sizes. The kinetic diameter of $CO_2$ and $C_2H_2$ (3.3 Å) was one of the smallest values in common gases, which caused a great obstacle in controlling the diffusion using porous materials for kinetic discrimination[20]. Nevertheless, **FDC−3a** could control the diffusion process of both $CO_2$ and $C_2H_2$ and amplify their slight rate difference, thus causing substantial differences in $T_{max}$ and achieving temperature-switched selective adsorption of $CO_2$ and $C_2H_2$.

## PXRD and solid-state NMR analyses

To further understand the mechanism from a structural aspect, in-situ PXRD patterns were collected during the adsorption process. Patterns obtained during the adsorptions of $CO_2$ (200 K) and $C_2H_2$ (300 K) did not reveal any structural changes (Supplementary Figs. 18, 19). We further performed synchrotron variable-temperature powder X-ray diffraction (VT-PXRD) measurements for **FDC−3a** from 100 to 380 K, which revealed that several peaks slightly shifted to lower angles upon increasing temperature (Supplementary Fig. 20). Taking the peak corresponding to the (111) plane as an example, it shifted from $2\theta = 5.079°$ to 5.015° as the temperature increased from 100 to 380 K,

which revealed the expansion of the [111] axis. Because one OPTz moiety was oriented parallel to the (111) plane (Supplementary Fig. 21), this slight expansion could be related to the extent of thermal flipping, which enlarged the gates to allow gas diffusion and controlled the diffusion rate.

Although the PXRD analysis could characterize the lattice change at different temperatures in **FDC−3a**, it could not show the local motions at the molecular level. To precisely reveal the flip-flop motion in **FDC−3a**, we performed $O_2$-enhanced high-resolution solid-state $^{13}$C cross-polarization magic angle spinning (CPMAS) nuclear magnetic resonance (NMR) study[21], which found around 25 well-distinguished sharp resonances including 11 quaternary and 15 tertiary aromatic carbon signals (Supplementary Figs. 22, 23), corresponding to the 32 total carbons (11 quaternary and 21 tertiary) of the ligand (Fig. 3a, Supplementary Table S6), in comparison with the solution $^{13}$C NMR attribution of the OPTz-t3da ligand based on calculated chemical shifts (Supplementary Fig. 24). This demonstrated the chemical purity and crystallinity of **FDC−3a** without observable defects. By contrast, the solution $^{13}$C NMR spectrum of OPTz-t3da ligand showed 17 distinct signals that are attributed to 17 chemically inequivalent carbons of the symmetric ligand (Supplementary Fig. 25). This contrast unambiguously revealed that the coordinated ligand in the framework is asymmetric, especially the two benzoate moieties (two distinct sets of $^{13}$C

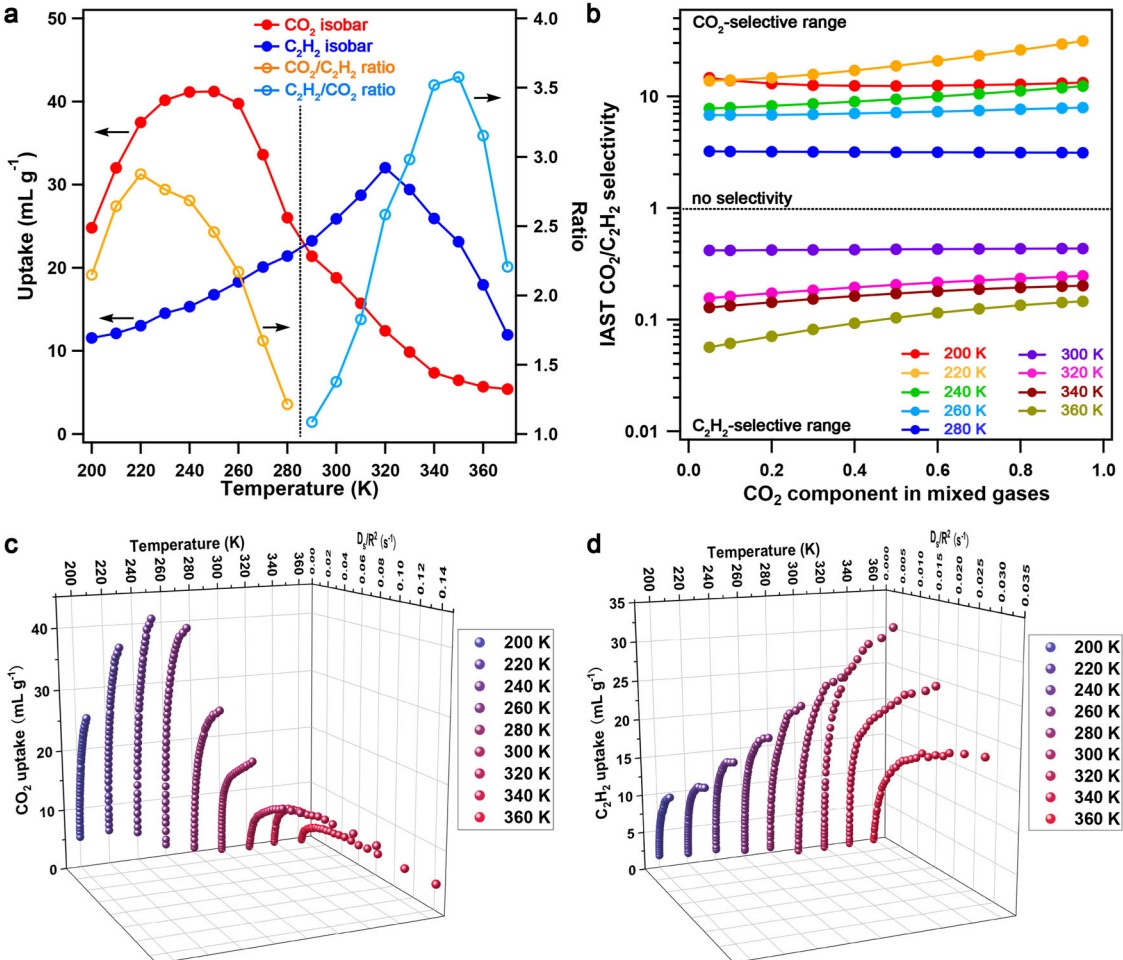

**Fig. 2 | Gas adsorption behavior, IAST selectivities, and diffusion rates between 200 and 370 K. a** $CO_2$ and $C_2H_2$ adsorption isobars at 1 bar, and the $CO_2/C_2H_2$ and $C_2H_2/CO_2$ uptake ratios. **b** IAST selectivities of **FDC−3a** for $CO_2/C_2H_2$ with different feed-gas components at various temperatures. **c** Global temperature−diffusion-rate−adsorption amount ($T–D_s/R^2–V$) landscape for $CO_2$ adsorption, where $R$ denotes the radius of an **FDC−3a** particle. **d** Global temperature−diffusion-rate−adsorption amount ($T–D_s/R^2–V$) landscape for $C_2H_2$ adsorption, where $R$ denotes the radius of an **FDC−3a** particle.

resonances for C1-C7), while the two rings of OPTz are nearly symmetric (single or two close resonances for each of C12-C17). Such asymmetry suggests rigid benzoate coordination with two distinctive environments and flexible OPTz with two similar rings, which was fully consistent with the asymmetric unit of the crystallographic structure containing monodentate and bidentate carboxylates. Moreover, the variable-temperature (VT) $^{13}C$ CPMAS NMR study from 213 to 352 K (Fig. 3a, Supplementary Fig. 25) found that the resonances of C12, C13, C16, and C17 on the OPTz ring shifted by 0.3–0.8 ppm while the resonances of C14 and C15 remained almost unchanged, indicative of the partial configurational change and reoriented OPTz rings in the ligand, which was in good correlation with the finding of slight lattice expansion by VT-PXRD as well as with the thermal flipping of OPTz moiety.

## Theoretical calculations

To get insight into the diffusion and adsorption process, Monte Carlo simulations and density functional theory (DFT) calculations were initially employed to optimize the adsorption positions of $CO_2$ and $C_2H_2$ molecules in **FDC−3a**. The optimized cell parameters for the activated phase are barely different from those of experimental values of **FDC−3a** (Supplementary Fig. 26, Supplementary Table 7), suggesting the reliability of the optimized crystal structure of **FDC−3a**. In the optimization of $CO_2$- and $C_2H_2$-adsorbed structures, cell parameters were kept the same as those of empty **FDC−3a**, because the adsorption

of $CO_2$ and $C_2H_2$ induced little structural transformation (Supplementary Fig. 10). Based on the above theoretical model, we investigated the $CO_2$ and $C_2H_2$ adsorption and diffusion (Fig. 3, b, c). The $CO_2$ and $C_2H_2$ diffusion barriers in **FDC−3a** were 35.8 and 55.9 kJ mol$^{-1}$, respectively (Supplementary Fig. 27a). The large diffusion barrier differences indicated that the diffusion of $CO_2$ was more kinetically favorable than the diffusion of $C_2H_2$. Indeed, although a self-accelerated adsorption process was demonstrated as a result of the temperature-promoted diffusion coefficients, the relative diffusion coefficient of $CO_2$ was perpetually higher than $C_2H_2$ by $7 \times 10^5$- and 400-folds at low and high temperatures, respectively (Supplementary Fig. 27b). Such a large difference in diffusion rate at low temperatures suggests that the adsorption of $C_2H_2$ is much farther from equilibrium than that of $CO_2$, resulting in its smaller adsorption amount than that of $CO_2$ at low temperatures despite that the adsorption energy (−40.4 kJ mol$^{-1}$) of $C_2H_2$ is stronger than that (−26.5 kJ mol$^{-1}$) of $CO_2$ (Supplementary Table 7). On the other hand, both $C_2H_2$ and $CO_2$ adsorptions can reach adsorption equilibrium at high temperatures despite their different diffusion rates, leading to the selective adsorption of $C_2H_2$ over $CO_2$ at high temperatures.

## Mixed gas separation

The temperature-switched adsorption behavior and its diffusion-regulatory mechanism in **FDC−3a** inspired us to perform dynamic mixed gas separation experiments; these were carried out with

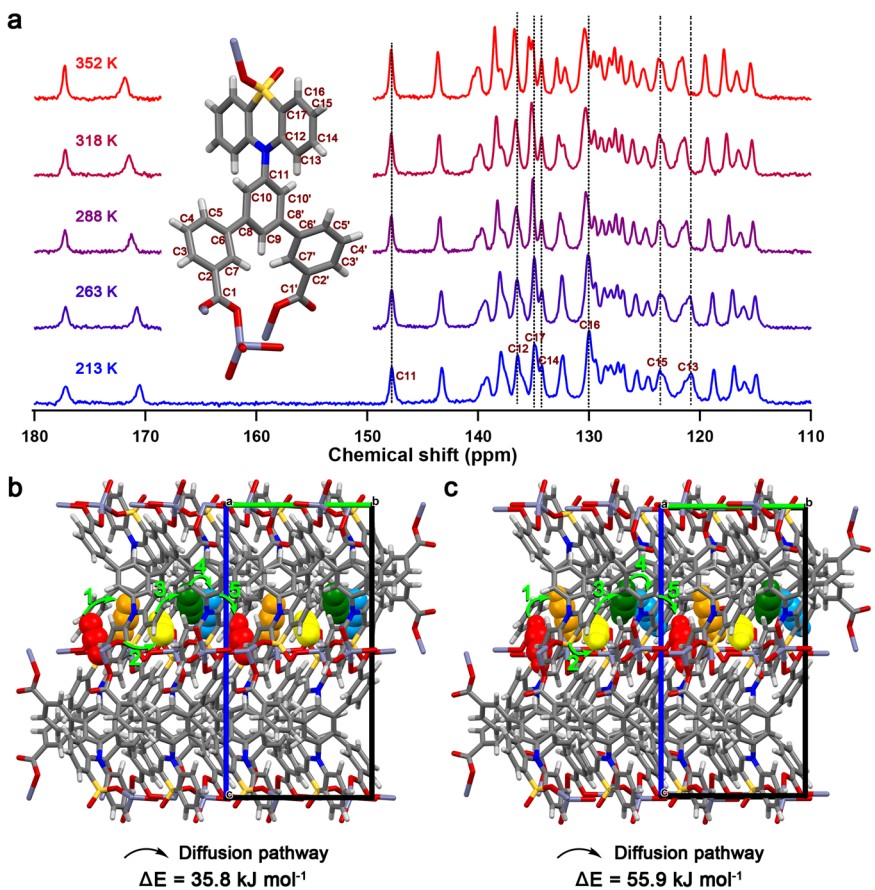

**Fig. 3 | Variable-temperature solid-state NMR and theoretical study. a** VT $^{13}$C CPMAS NMR spectra of **FDC−3a**. **b** Simulated structure of the $CO_2$-adsorbed phase and schematic diagram of diffusion pathways of **FDC−3a**, where $CO_2$ molecules outside the representative path are omitted for clarity. The $CO_2$ molecules at the initial position are marked with red color, and the $CO_2$ molecules at the other positions of the diffusion pathway are marked with orange, yellow, green, and cyan colors, respectively. Numbers 1 to 5 and the arrows represent the diffusion pathways. **c** Simulated structure of the $C_2H_2$-adsorbed phase and schematic diagram of diffusion pathways of **FDC−3a**, where $C_2H_2$ molecules outside the representative path are omitted for clarity. The $C_2H_2$ molecules at the initial position are marked with red color, and the $C_2H_2$ molecules at the other positions of the diffusion pathway are marked with orange, yellow, green, and cyan colors, respectively. Numbers 1 to 5 and the arrows represent the diffusion pathways.

temperature-programmed desorption (TPD) protocol (Supplementary Figs. 28, 29)[18,19]. Considering the adsorption amounts and the selectivity, we conducted the separation experiments at 240 and 320 K, respectively. At the low temperature of 240 K, **FDC−3a** selectively adsorbed $CO_2$ from a nearly equimolar $CO_2/C_2H_2$ mixture ($CO_2$:$C_2H_2$ = 54.2:45.8) within a short exposure time of 1 h, leading to a remarkable $CO_2$ enrichment with a composition up to 97.7% in the adsorbed phase (Fig. 4a, Supplementary Fig. 34) and a $CO_2/C_2H_2$ separation factor of 36 (Fig. 4b). The separation factor was comparable with the IAST selectivity, though the values of the former were larger than the latter. The high $CO_2/C_2H_2$ separation factor indicated that $CO_2$ diffused much faster than $C_2H_2$, thus occupying most of the available space and excluding the $C_2H_2$ by a molecular-sieving mechanism. **FDC−3a** exhibited outstanding $CO_2$ enrichment over a wide range of feed-gas components (Supplementary Figs. 30–40); even though the mixture was in a composition of $CO_2$:$C_2H_2$ = 4.0:96.0, **FDC−3a** enriched $CO_2$ resulting in a $CO_2$ concentration of 95.4% in the adsorbed phase (Fig. 4a) and a $CO_2/C_2H_2$ separation factor of 498. On the other hand, At the high temperature of 320 K, **FDC−3a** selectively adsorbed $C_2H_2$ from a nearly equimolar $CO_2/C_2H_2$ mixture ($CO_2$:$C_2H_2$ = 54.2:45.8) within a short exposure time of 1 h, leading to a remarkable $C_2H_2$ enrichment with a composition up to 94.1% in the adsorbed phase (Fig. 4a, Supplementary Fig. 45) and a $CO_2/C_2H_2$ separation factor of $5.4 \times 10^{-2}$ (i.e., $C_2H_2/CO_2$ separation factor of 18, Fig. 4b). The high $C_2H_2/CO_2$ separation factor suggested that **FDC−3a**

was thermodynamically favorable to $C_2H_2$ when the diffusion-regulatory mechanism did not work at high temperatures. **FDC−3a** exhibited marked $C_2H_2$ enrichment over a wide range of feed-gas components (Supplementary Figs. 41–51); even though the mixture was in a composition of $CO_2$:$C_2H_2$ = 93.8:6.2, the $C_2H_2$ concentration was 92.6% in the adsorbed phase (Fig. 4a), which corresponds to a $CO_2/C_2H_2$ separation factor of $5.5 \times 10^{-3}$ (i.e., $C_2H_2/CO_2$ separation factor of 181, Fig. 4b). Notably, the selectivities of the mixed-gas separation were higher than the ones predicted from the single-gas adsorption. At low temperatures, the diffusion of both $CO_2$ and $C_2H_2$ was regulated, and the gas separation was governed by the diffusion-rate difference. $CO_2$ showed a faster diffusion rate than $C_2H_2$, resulting in high $CO_2/C_2H_2$ selectivities under non-equilibrium states. On the other hand, at high temperatures, the cooperativity of $CO_2$-$C_2H_2$ interaction and gas-framework interaction amplified the selective adsorption of $C_2H_2$, rendering a higher $C_2H_2/CO_2$ selectivity than the expected one from the single-gas adsorption.

## Discussion

Our findings provide temperature-switched recognition of the $CO_2$ and $C_2H_2$ by controlling their diffusion and amplifying the rate differences. The TPD results for kinetic gas separation of $CO_2/C_2H_2$ binary mixtures demonstrate temperature-dependent high selectivities with a $CO_2/C_2H_2$ separation factor of 498 at 240 K and a $C_2H_2/CO_2$ separation factor of 181 at 320 K. These striking separation features should give

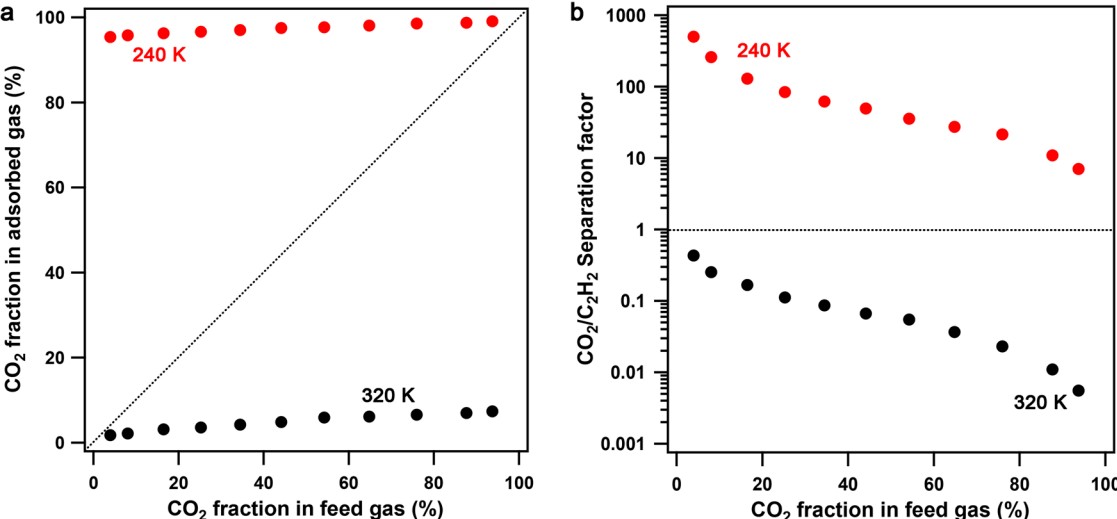

**Fig. 4 | Mixed gas separation. a** McCabe-Thiele diagram for $CO_2/C_2H_2$ separation by **FDC−3a** at 240 and 320 K, with the dashed line representing the theoretical behavior of a material showing no selectivity. **b** The correlation between $CO_2$ concentration in the feed gas and the $CO_2/C_2H_2$ separation factor.

the credit to the underlying mechanism, which is implemented by the cooperation of ultrasmall pore apertures and local dynamics of gate constituents. This design principle can be extensively adaptable with various host-guest systems for manipulatable selectivity trends by external stimuli for recognizing similar guests.

## Methods

### Synthesis of FDC−3
Firstly, 50 mg (0.09 mmol) **OPTz-t3da** was dissolved in 2 mL DMA at room temperature. A methanol solution (8 mL) of $Zn(NO_3)_2 \cdot 6H_2O$ (54 mg, 0.18 mmol) was added to the above solution. Then the mixture was heated at 80 °C for 72 h. **FDC−3** was obtained as colorless block crystals with sizes up to several hundreds of micrometers (37 mg, yield = 43%). The crystals were filtered, washed with DMA (10 mL, 3 times) and methanol (10 mL, 3 times), and dried in air. The as-synthesized **FDC−3** was characterized by infrared spectra (Supplementary Fig. 3). The adsorption peak of the stretching vibration of the C = O double bond shifted to a low wavenumber, indicative of the coordination bond formation in **FDC−3**.

### Solvent exchange and activation of FDC−3
To measure the adsorption property of **FDC−3**, we exchanged the guest and coordination solvents (DMA) with methanol by soaking **FDC−3** in methanol at 60 °C for 7 days. Every 24 h the methanol was replaced by a new one. After the solvent exchange, the exchanged **FDC−3** was dried under vacuum at 60 °C for 3 h. $^1H$ NMR confirmed that the DMA in the exchanged **FDC−3** was exchanged by methanol (Supplementary Fig. 8).

TG curve showed that the framework of the exchanged **FDC−3** was thermally stable until 391 °C, whereas below 60 °C the exchanged **FDC−3** lost the methanol molecules (Supplementary Fig. 9). Thus, we activated the exchanged **FDC−3** at 120 °C for 11 h to afford **FDC−3a**; this temperature ensured the complete removal of the solvents meanwhile excluding the possibility of framework decomposition.

## Data availability
The data that support the plots within this paper and other finding of this study are available from the corresponding authors upon reasonable request. Source data are provided in this paper. The X-ray crystallographic coordinates for structures reported in this study have been deposited at the Cambridge Crystallographic Data Centre (CCDC), under deposition numbers 2236266-2236267. These data can be obtained free of charge from The Cambridge Crystallographic Data Centre via www.ccdc.cam.ac.uk/data_request/cif. Source data are provided with this paper.

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

## Acknowledgements

This work was supported by the National Natural Science Foundation of China (21975078), the Fundamental Research Funds for the Central Universities, the start-up foundation of Sichuan University, the KAKENHI Grant-in-Aid for Specially Promoted Research (JP25000007), and Scientific Research (S) (JP18H05262) from the Japan Society of the Promotion of Science (JSPS). We thank the iCeMS analysis center for access to the analytical instruments.

## Author contributions

Y.S. performed experiments associated with molecular synthesis, crystal growth, gas sorption, and gas separation. K.O. and P.W. conducted single-crystal and powder XRD studies and structure analyses. J.Z. and H.X. carried out calculation studies. Q.W. and H.L. conducted solid-state NMR measurements. F.H. performed cRED measurements and solved the structure of the activated phase. C.G. and S.K. conceived the project and directed the research. All authors contributed to the writing and editing of the manuscript.

## Competing interests

The authors declare no competing interests.
