## [Peer review file · Nature Communications]

REVIEWER COMMENTS

Reviewer #1 (Remarks to the Author):

The authors report on an interesting observation, namely the temperature-dependent selectivity switch from CO₂ to C₂H₂ using a small-pore MOF as adsorbent. As the origin of this selectivity change they assume a flip-flop motion of a unit of the linker molecule used for the preparation of the MOF adsorbent. I do concede that the authors have an interesting finding but I do not think that the interpretation of the data is correct.

The flipping motion is considered to proceed via a phenothiazine-5,5-dioxide (OPTz) unit. The authors have calculated that the rotation may proceed as the energy barrier is only 25 kJ/mol. This value holds for the OPTz unit in the as-made MOF. Unfortunately, the MOF undergoes a structural reorganisation during activation, resulting in oxygen atoms from the SO₂ group of the OPTz coordinating the zinc nodes. Consequently, a free rotation of the OPTz is inhibited. If such a motion would proceed, bond-breaking and bond formation would be required which is not comparable to the process calculated for the freely rotating OPTz. Thus, the proposed flip-flop mechanism is likely not possible.

As an argument for such a motion, the authors report a structural expansion during temperature increase. Such thermal expansion is a very common feature of basically all crystal structures, simply due to stronger thermal motion of the atoms upon temperature increase.

Thus, there are no real arguments that would hold in favour of the proposed mechanism. The selectivities observed are likely triggered by the interplay of different diffusivity and adsorptive interaction of the molecules, as also assumed by the authors. Their MD calculation with a rigid MOF framework support that notion.

I am also a bit sceptical about the structural data provided. Single crystal investigation is performed at one individual crystallite, chosen at random. In contrast powder diffraction probes a large ensemble of crystallites simultaneously. If the structure determined by single crystal diffraction is representative for the complete sample, then the theoretical powder diffraction pattern, calculated from the single crystal structure data, should be identical with the measured powder diffraction data. Reflection intensities may show slight differences, but the 2θ positions of the reflections must be identical. This is definitely not the case for the materials reported here (see Fig. S10), indicating that the real structure of the MOF is not represented by the single crystal data. As the consequence, the structural data must be questioned.

As a minor point, the authors should inspect the equation for the separation factor in the SI. The first term reported there is obviously wrong.

In summary, I do not think that the present manuscript is suitable for publication for the reasons given above. I therefore recommend its rejection. The findings by the authors are still interesting and I would encourage the authors to continue the work on the subject.

Reviewer #2 (Remarks to the Author):

This paper describes a marked temperature dependence of gas adsorption on PCP due to entrance-blocking pores. The drastic selectivity change with temperature is quite interesting, but not surprising with the relevance to the competitive adsorption and diffusion around the pore entrances. Very small

pore materials such as coal and carbon molecular sieves often shows the reverse adsorption behavior with changing the adsorption temperature. For example, A book written by H. Marsh and F. Rodriguez-Reinoso, *Activated Carbon* ((2006) Elsevier) describes the figure of the adsorption amount vs. adsorption temperature relationship. However, surely such reverse temperature dependence of adsorption should be firstly observed in the PCP systems because of new porous materials. Also, it is notable that the adsorption selectivity change is evidently shown. This is publishable. The crystal structure of PCP can be correctly understood through the crystal structure, being quite different from the coal and carbon molecular sieves. Therefore, the authors are expected to give the precise temperature dependence mechanisms which is associated with the entrance blocking. A more quantitative discussion for the reverse mechanism of the adsorption selectivity is preferable for publication. Authors could evaluate theoretically the adsorption selectivity dependence on adsorption for rigorous discussion for publication.

The followings are just comments for the manuscript.

1. Authors studied CO₂ and C₂H₂ which have almost similar molecular geometries. This is quite nice. Furthermore, authors could consider electronic properties difference in addition to the geometrical factor (dispersion interaction and quadrupole moment associated interaction). The π electronic factor in C₂H₂ may be important. Also, MIN-2 parameters may be introduced for discussion, even though their difference is tiny.
2. The adsorption irreversibility may be described.
3. Authors use "affinity" for matching with "recognition". However, those concepts are not clear compared with selective adsorption. As the reason for recognition comes from the in-pore diffusion barrier, ambiguous expressions may be avoided.
4. The description: separation factor of 498.1--- may be correctly described. The reproducibility of adsorption cannot have such a large significant number. 498 or 500 ?
5. Authors may describe the absolute diffusion rate which can be preferable to compare with the data of other porous materials.
6. "The temperature-assisted adsorption behavior could be considered as a kinetic factor" on page 5 is not clear.
7. Authors evaluated the adsorption selectivity using IAST theory and measured the mixed gas adsorption. However, they do not discuss their difference for better understanding the gas-PCP interaction.
8. " it could not show the local motions at the molecular level" on page 7. The expression may be improved.
9. Authors can show the extent of expansion due to the peak shift of XRD on page 7.
10. MC simulations are not necessarily fit for diffusion process. Probably authors should have the MD data. The energy barriers are so large compared with the thermal energies, although authors discuss the adsorption selectivity using these energy barriers.

Reviewer #3 (Remarks to the Author):

The manuscript reported the adsorption and diffusion of CO₂ and C₂H₂ in a flexible MOF, and revealed an interesting switching molecular recognition selectivities by temperature. The obtained results may provide useful information for manipulatable selectivity trends by external stimuli for recognizing similar guests. However, considering the following issues, the current state of this manuscript is not suitable for publication in Nature Communications.

1. The authors should provide the PXRD patterns of the FDC-3 loaded with CO₂ and C₂H₂ guest molecules at different pressures, to demonstrate that the stimulus response of the material is only related to temperature.
2. The author takes the separation of CO₂ and C₂H₂ as an example, thus the corresponding breakthrough experiments and the physicochemical stability of the material should also be further characterized to confirm its application in practical industry.
3. In Figure S9, why does the activated material still have a mass loss before 100 °C? Does the activated material adsorb other molecules in the air?
4. In Figure S10, the simulated PXRD patterns of the sample do not correspond well to that of the experimental results, and the authors should confirm the correctness of the structure. Why do the PXRD patterns in Figure S10 and S17 in SI have different diffraction peaks between 4-12° for FDC-3a?
5. The authors have only explained the slight shift of the diffraction peak 5.079-5.015° in the variable temperature PXRD pattern. While the diffraction peak around 12° in Figure S17 is significantly shifted, why?
6. In this paper, the authors focus on the adsorption or diffusion of guest molecules in temperature-responsive FDC-3, and only illustrate by variable-temperature XRD and solid-state NMR that the ligands in the temperature-elevated FDC-3a are flipped, which affects the adsorption of guest molecules on FDC-3a. It is suggested to provide pure-component static and kinetic adsorption experiments of the material at different temperatures to further demonstrate the experimental phenomenon.
7. The author described in the manuscript that the pore size of the material is 2.9 Å, however, it can adsorb CO₂ (3.3 Å) and C₂H₂ (3.3 Å) at relatively high temperatures, indicating a change in the pore size of the material. Based on this, the single component adsorption isotherms at lower temperatures and the pore size of material at different temperatures should also be presented in the manuscript.
8. When calculating the IAST selectivity, the corresponding fitting model, fitting parameters, and correlation coefficients are missing.
9. According to the previous article (Nature Communications, 2023, 14, 3789), is it possible that the guest molecules form tetramers in the pores when the material adsorbs guest molecules at low temperatures? What is the packing density? Is the force dominated by the guest-guest or host-guest interactions?
10. For the theoretical calculations, how to confirm the reliability of the Universal force field in MC simulations? The details for calculating the diffusion pathways should be provided. For simulating diffusion, it is reasonable to use the supercell model instead of a unit cell, since the pore aperture plays a significant effect.

Response to Referee #1

The authors report on an interesting observation, namely the temperature-dependent selectivity switch from CO₂ to C₂H₂ using a small-pore MOF as adsorbent. As the origin of this selectivity change they assume a flip-flop motion of a unit of the linker molecule used for the preparation of the MOF adsorbent. I do concede that the authors have an interesting finding but I do not think that the interpretation of the data is correct.

We appreciate the comments from the reviewer endorsing the interest of our work. Regarding the concern raised by the reviewer, we have taken significant consideration of all the comments and suggestions from the reviewer, performed additional experiments, and revised our manuscript accordingly. We believe that the revised manuscript has clarified all the requests and comments from the reviewer.

The flipping motion is considered to proceed via a phenothiazine-5,5-dioxide (OPTz) unit. The authors have calculated that the rotation may proceed as the energy barrier is only 25 kJ/mol. This value holds for the OPTz unit in the as-made MOF. Unfortunately, the MOF undergoes a structural reorganisation during activation, resulting in oxygen atoms from the SO₂ group of the OPTz coordinating the zinc nodes. Consequently, a free rotation of the OPTz is inhibited. If such a motion would proceed, bond-breaking and bond formation would be required which is not comparable to the process calculated for the freely rotating OPTz.

Thus, the proposed flip-flop mechanism is likely not possible.

As an argument for such a motion, the authors report a structural expansion during temperature increase. Such thermal expansion is a very common feature of basically all crystal structures, simply due to stronger thermal motion of the atoms upon temperature increase.

Thus, there are no real arguments that would hold in favour of the proposed mechanism. The selectivities observed are likely triggered by the interplay of different diffusivity and adsorptive interaction of the molecules, as also assumed by the authors. Their MD calculation with a rigid MOF framework support that notion.

We thank the reviewer for this thought-provoking comment. To defend our claim of the flip-flop mechanism, we performed the theoretical calculation of the flip-flop energy of the

ligand by constraining the O atoms on the OPTz and carboxyl moieties to simulate the ligand in the activated PCP. The flip-flop energy was similar to the free ligand when the flip-flop angle was within $\pm 20^\circ$ (less than 25 kJ mol^{-1}). This result demonstrated that the flip-flop motion was essentially feasible even in the activated PCP, without the request of bond-breaking and bond-formation processes as proposed by the reviewer. Indeed, this could also be proved by solid-state NMR and synchrotron PXRD results in our manuscript.

Of course, we agree with the reviewer that the interplay of different diffusivity and adsorptive interactions of the molecules played a role in the observed selective adsorption. However, without the flip-flop mechanism, the pore entrance is smaller than the kinetic diameters of CO_2 and C_2H_2 , and the gas molecules were unlikely to be adsorbed because of the large energy barriers at the pore entrance. Therefore, the flip-flop motion was the essence and prerequisite to achieving the observed selectivity.

To clearly show the flip-flop energies in the different states, we have added the new calculation results to the revised Supplementary Materials as Fig. S1b.

Supplementary Figure S1. (a) Potential energy surface for the flip-flop of OPTz ring in free ligand molecules. (b) Potential energy surface for the flip-flop of OPTz ring in oxygen atom-constrained ligand molecules to simulate the activated PCP. The energy change (ΔE) for the OPTz ring flipping is less than 25 kJ mol^{-1} with respect to a change of 25° in the dihedral angle. Therefore, the thermal flipping of the OPTz ring is feasible.

I am also a bit sceptical about the structural data provided. Single crystal investigation is performed at one individual crystallite, chosen at random. In contrast powder diffraction probes

a large ensemble of crystallites simultaneously. If the structure determined by single crystal diffraction is representative for the complete sample, then the theoretical powder diffraction pattern, calculated from the single crystal structure data, should be identical with the measured powder diffraction data. Reflection intensities may show slight differences, but the 2θ positions of the reflections must be identical. This is definitely not the case for the materials reported here (see Fig. S10), indicating that the real structure of the MOF is not represented by the single crystal data.

As the consequence, the structural data must be questioned.

We apologize for causing confusion to the reviewer. Indeed, the single-crystal measurements by electron diffraction were reproduced several times, yielding the same structure as we reported. Regarding the PXRD, it was measured by the staff of our institute, and we were not able to perform the measurements by ourselves. When we send samples to him, he does not start the measurement right away. As the climate of our city is perennially humid, the samples easily adsorb moisture and cause structural transformation. Therefore, to solve this problem, we performed *in-situ* gas sorption/PXRD measurements, where the sample was activated *in situ* in the chamber of the machine. Using this method, we obtained the PXRD pattern of the activated phase, which matched well with the PXRD pattern calculated from the single-crystal structure, and further demonstrated the accuracy of the single-crystal structure by electron diffraction.

According to the suggestion from the reviewer, we have replaced the original Fig. S10 with a new one in the revised Supplementary Materials.

Supplementary Figure S10. PXRD patterns of **FDC-3** (simulated and experimental) and **FDC-3a** (simulated and experimental). The PXRD pattern of as-synthesized **FDC-3** was different from the simulated one because of the structural change upon the loss of volatile solvents (water and methanol) in the crystal-collection process.

As a minor point, the authors should inspect the equation for the separation factor in the SI. The first term reported there is obviously wrong.

We are sorry for the typo. We have revised the equation of the separation factor in the revised Supplementary Materials.

In summary, I do not think that the present manuscript is suitable for publication for the reasons given above. I therefore recommend its rejection.

The findings by the authors are still interesting and I would encourage the authors to continue the work on the subject.

According to the suggestion from the reviewer, we have performed additional experiments and revised our manuscript critically. We believe the revision has clarified all the issues raised by the reviewer. We sincerely hope the reviewer can understand our positive situation regarding

the revision and be satisfied with the additional experiments and revisions we performed. We appreciate the reviewer for the above constructive comments that improve the manuscript.

=====

Response to Referee #2

This paper describes a marked temperature dependence of gas adsorption on PCP due to entrance-blocking pores. The drastic selectivity change with temperature is quite interesting, but not surprising with the relevance to the competitive adsorption and diffusion around the pore entrances. Very small pore materials such as coal and carbon molecular sieves often shows the reverse adsorption behavior with changing the adsorption temperature. For example, A book written by H. Marsh and F. Rodriguez-Reinoso, Activated Carbon ((2006) Elsevier) describes the figure of the adsorption amount vs. adsorption temperature relationship. However, surely such reverse temperature dependence of adsorption should be firstly observed in the PCP systems because of new porous materials. Also, it is notable that the adsorption selectivity change is evidently shown. This is publishable.

The crystal structure of PCP can be correctly understood through the crystal structure, being quite different from the coal and carbon molecular sieves. Therefore, the authors are expected to give the precise temperature dependence mechanisms which is associated with the entrance blocking. A more quantitative discussion for the reverse mechanism of the adsorption selectivity is preferable for publication. Authors could evaluate theoretically the adsorption selectivity dependence on adsorption for rigorous discussion for publication.

=====

We appreciate the comments from the reviewer endorsing our work. Regarding the concern raised by the reviewer, we have taken significant consideration of all the comments and suggestions from the reviewer and revised our manuscript accordingly. We believe that the revised manuscript has clarified all the requests and comments from the reviewer.

=====

The followings are just comments for the manuscript.

1. Authors studied CO₂ and C₂H₂ which have almost similar molecular geometries. This is quite nice. Furthermore, authors could consider electronic properties difference in addition to the geometrical factor (dispersion interaction and quadrupole moment associated interaction). The π electronic factor in C₂H₂ may be important. Also, MIN-2 parameters may be introduced for discussion, even though their difference is tiny.

=====

According to the suggestion from the reviewer, we have added the polarizabilities (positively related to the dispersion interaction) and quadrupole moments to Table S3 in the revised Supplementary Materials. We also revised one sentence in the manuscript:

“Remarkably, the T_{max} values of CO_2 and C_2H_2 were largely different by 80 K, although they had exactly the same kinetic diameters and very similar molecular sizes and polarizabilities (Supplementary Table S3).”

As for the MIN-2 parameters, we are so sorry that we were not able to know what this is, even though we have surveyed in literature and consulted the experts in gas sorption and theoretical calculations. Therefore, we apologize that we are not able to discuss this in the current version of the manuscript. We thank the reviewer if he/she could provide some related literature, and we will add the discussion accordingly.

2. The adsorption irreversibility may be described.

In Fig. S12 in the Supplementary Materials, obvious desorption hysteresis was observed for CO_2 (200 to 300 K) and C_2H_2 (200 to 360 K) in their isotherms. According to our previous study (*Science* **2019**, 363, 387), this desorption hysteresis was characteristic of the diffusion-regulatory pore systems in PCPs. These results further indicated that the diffusions of CO_2 and C_2H_2 were regulated in the temperature ranges of 200 to 300 K and 200 to 360 K, respectively, showing that the adsorption of CO_2 was controlled by kinetics and thermodynamics at low (200 to 300 K) and high (320 to 360 K) temperatures, respectively, whereas the adsorption of C_2H_2 was constantly controlled by kinetics.

According to the suggestion from the reviewer, we have added several sentences in the revised manuscript to describe the desorption hysteresis in the isotherms:

“Additionally, obvious desorption hysteresis was observed for CO_2 (200 to 300 K) and C_2H_2 (200 to 360 K) in their sorption isotherms, which was characteristic of the diffusion-regulatory pore systems in PCPs^{17,18}. These results further indicated that the diffusions of CO_2 and C_2H_2 were regulated in the temperature ranges of 200 to 300 K and 200 to 360 K, respectively, showing that the adsorption of CO_2 was controlled by kinetics and thermodynamics at low (200 to 300 K) and high (320 to 360 K) temperatures, respectively, whereas the adsorption of C_2H_2 was constantly controlled by kinetics.”

3. Authors use “affinity” for matching with “recognition”. However, those concepts are not clear compared with selective adsorption. As the reason for recognition comes from the in-pore diffusion barrier, ambiguous expressions may be avoided.

We appreciate the reviewer for this comment. Accordingly, we have replaced the word “*recognition*” with “*selective adsorption*” in the revised manuscript.

4. The description: separation factor of 498.1--- may be correctly described. The reproducibility of adsorption cannot have such a large significant number. 498 or 500 ?

According to the suggestion from the reviewer, we have revised all the significant numbers of the separation factors in the revised manuscript.

5. Authors may describe the absolute diffusion rate which can be preferable to compare with the data of other porous materials.

We appreciate the reviewer for this comment. However, in the Crank theory, the diffusion rate is always accompanied by the R parameter, namely, the radius of a PCP particle. As one is not able to measure all the radii of all the PCP particles, the diffusion rate calculated by the Crank theory is an average value in a statistical sense. Therefore, it is imprecise to directly compare the diffusion rate of our system to other literature. Nevertheless, we could find some related systems using the Crank theory to calculate the diffusion rate. For instance, in the paper *Nature* **2021**, 595, 542, D. Li *et al.* used the Crank theory to calculate the diffusion rates of C_3H_6 in three MOFs, JNU-3a, KAUST-7, and Y-abtc, which exhibited the diffusion rates (D) of 2.94×10^{-3} , 8.66×10^{-4} , and $7.64 \times 10^{-5} R^2 s^{-1}$, respectively. These values were comparable with ours (3.22×10^{-2} and $8.29 \times 10^{-3} R^2 s^{-1}$ for CO_2 and C_2H_2 at 240 K; 7.24×10^{-2} and $2.18 \times 10^{-2} R^2 s^{-1}$ for CO_2 and C_2H_2 at 320 K), though the direct comparison of CO_2 , C_2H_2 , and

C₃H₆ is difficult as the diffusion rates of CO₂ and C₂H₂ are substantially higher than C₃H₆ in their gaseous states.

6. “The temperature-assisted adsorption behavior could be considered as a kinetic factor” on page 5 is not clear.

According to the suggestion from the reviewer, we have replaced the original sentence with a new one “*The temperature-assisted adsorption behavior was controlled by kinetics...*” in the revised manuscript.

7. Authors evaluated the adsorption selectivity using IAST theory and measured the mixed gas adsorption. However, they do not discuss their difference for better understanding the gas-PCP interaction.

We thank the reviewer for this comment. The IAST selectivities of CO₂/C₂H₂ and C₂H₂/CO₂ for an equimolar mixture were 18.6 and 9.5 at 220 and 360 K, respectively. By contrast, the separation factors for the same gas mixture were 36 (220 K) and 18 (360 K), respectively. The separation factors were comparable with the IAST selectivities, though the values of the former were larger than the latter.

According to the suggestion from the reviewer, we have added one sentence to the revised manuscript to discuss the separation factors and the IAST selectivities:

“The separation factor was comparable with the IAST selectivity, though the values of the former were larger than the latter.”.

We sincerely hope that the reviewer can understand our favorable attitude regarding the revision and be satisfied with the revision we made. We appreciate the reviewer for the above constructive comments that improve the manuscript.

Response to Referee #3

The manuscript reported the adsorption and diffusion of CO₂ and C₂H₂ in a flexible MOF, and revealed an interesting switching molecular recognition selectivities by temperature. The obtained results may provide useful information for manipulatable selectivity trends by external stimuli for recognizing similar guests. However, considering the following issues, the current state of this manuscript is not suitable for publication in Nature Communications.

=====

We appreciate the comments from the reviewer endorsing the interests of our work. Regarding the concern raised by the reviewer, we have taken significant consideration of all the comments and suggestions from the reviewer, performed additional experiments, and revised our manuscript accordingly. We believe that the revised manuscript has clarified all the requests and comments from the reviewer.

=====

1. The authors should provide the PXRD patterns of the FDC-3 loaded with CO₂ and C₂H₂ guest molecules at different pressures, to demonstrate that the stimulus response of the material is only related to temperature.

=====

We appreciate the reviewer for this constructive comment. Accordingly, we performed *in-situ* gas sorption/PXRD measurements. The results showed that the PXRD patterns were consistent with the increased pressure, demonstrating that the stimuli response of **FDC-3a** is only related to temperature.

We have added the results to the revised Supplementary Materials as Figs. S17 and S18. We also added two sentences to the revised manuscript:

“To further understand the mechanism from a structural aspect, in-situ PXRD patterns were collected during the adsorption process. Patterns obtained during the adsorptions of CO₂ (200 K) and C₂H₂ (300 K) did not reveal any structural changes (Supplementary Figs. S17, S18).”

Supplementary Figure S17. Coincident *in-situ* adsorption/PXRD patterns of **FDC-3a** during CO₂ adsorption measured at 200 K at given equilibrium pressures.

Supplementary Figure S18. Coincident *in-situ* adsorption/PXRD patterns of **FDC-3a** during C₂H₂ adsorption measured at 300 K at given equilibrium pressures.

2. The author takes the separation of CO₂ and C₂H₂ as an example, thus the corresponding breakthrough experiments and the physicochemical stability of the material should also be further characterized to confirm its application in practical industry.

We agree with the reviewer that breakthrough experiments can evaluate the gas-separation potential of PCP materials for industrial applications. However, please allow us to explain a little bit why we did not include the breakthrough results in the manuscript. The essential

mechanism of our PCP for selective adsorption of CO₂ and C₂H₂ at different temperatures is diffusion regulation, namely, to slow down the diffusion in their gaseous phase to amplify their rate differences. By this mechanism, we have successfully achieved the separation of extremely challenging gases/vapors, such as O₂/Ar (*Science* **2019**, 363, 387) and H₂O/D₂O (*Nature* **2022**, 611, 289). However, because of similar gas-framework interactions of the above gases, we were not able to switch the selectivity at different temperatures. In this study, the diffusion of CO₂ was faster than C₂H₂, whereas the gas-framework interaction of CO₂ was smaller than C₂H₂, rendering the selective adsorption at different temperatures by the cooperativity of the kinetic and thermodynamic factors. Therefore, this manuscript provided a new mechanism and achievement even though we have published two papers in this field.

On the other hand, the diffusion regulation resulted in slow adsorption of the gases, whereas the breakthrough experiment was a dynamically rapid process within a short time scale. Therefore, although we have performed the breakthrough experiments, we were not able to separate CO₂ and C₂H₂ mixtures because the adsorption amount was too small in this short time. This is why we used the TPD protocol to evaluate the gas-separation performance as this method checked the ratio of the adsorbed gases.

Even though we were not able to provide more positive results in the breakthrough experiments, we still believe that our manuscript possesses significant novelty and the achievements in this manuscript deserve to be published in *Nat. Commun.* even without the supplement of the breakthrough results. This is because we realized the switch of molecular recognition selectivity by a new diffusion-regulatory mechanism, which was hardly achieved in traditional supramolecular systems. Notably, such a selectivity switch was realized by simply changing temperatures, in sharp contrast to the barely reported supramolecular systems that require complicated molecular design. Therefore, this manuscript focuses on basic sciences in the adsorption field, rather than the technical problem that can be accomplished by industrial experts. When a new mechanism or a new type of material appears, we should not treat it as if all the complimentary research has been finished. By contrast, we believe that a lot of research will be done to develop new PCP materials and related mechanisms to enrich the selectivity-switch systems. We think a significant ripple effect to selectivity-switchable mechanisms and PCP materials will be expected. Therefore, the results obtained here are not just technical but also scientific, including material design and its synthetic chemistry, which will contribute to the PCPs and supramolecular fields.

3. In Figure S9, why does the activated material still have a mass loss before 100 °C? Does the activated material adsorb other molecules in the air?

We apologize for causing confusion to the reviewer. The TG curves were measured by the staff of our institute, and we were not able to perform the measurements by ourselves. When we send samples to him, he does not start the measurement right away. As the climate of our city is perennially humid, the mass loss before 100 °C was because of the moisture adsorbed in the samples. A similar problem was also in the PXRD measurements (please see the response to the next comment).

4. In Figure S10, the simulated PXRD patterns of the sample do not correspond well to that of the experimental results, and the authors should confirm the correctness of the structure. Why do the PXRD patterns in Figure S10 and S17 in SI have different diffraction peaks between 4-12° for FDC-3a?

We apologize for causing confusion to the reviewer. The PXRD was measured by the staff of our institute, and we were not able to perform the measurements by ourselves. When we send samples to him, he does not start the measurement right away. As the climate of our city is perennially humid, the samples easily adsorb moisture and cause structural transformation. Therefore, to solve this problem, we performed *in-situ* gas sorption/PXRD measurements, where the sample was activated *in situ* in the chamber of the machine. Using this method, we obtained the PXRD pattern of the activated phase, which matched well with the PXRD pattern calculated from the single-crystal structure, and further demonstrated the accuracy of the single-crystal structure by electron diffraction.

According to the suggestion from the reviewer, we have replaced the original Fig. S10 with a new one in the revised Supplementary Materials.

Supplementary Figure S10. PXRD patterns of **FDC-3** (simulated and experimental) and **FDC-3a** (simulated and experimental). The PXRD pattern of as-synthesized **FDC-3** was different from the simulated one because of the structural change upon the loss of volatile solvents (water and methanol) in the crystal-collection process.

As for the difference of the PXRD patterns in Figs. S10 and S17, it was because we used the lab machine to record the PXRD patterns in Fig. S10 but employed the synchrotron PXRD facility to record the PXRD patterns in Fig S17 as the synchrotron could reflect the slight difference in varying temperatures. Because the wavelengths of the X-ray in the two measurements were 1.54178 and 0.80000 Å, respectively, according to the Bragg equation, the differences in the PXRD patterns are easily understood.

5. The authors have only explained the slight shift of the diffraction peak 5.079-5.015° in the variable temperature PXRD pattern. While the diffraction peak around 12° in Figure S17 is significantly shifted, why?

The peak around 12° in Figure S17 was the diffraction plane of (124), which was depicted in the **Reviewer-only Figure 1** (red plane). The “flipping” structure revealed a slight expansion of the [124] axis compared to the “static” structure, in good agreement with the VT-PXRD results that the peaks corresponding to the (124) facet shifted to a lower angle. Since one OPTz moiety is on the (124) plane and another OPTz moiety is parallel to the (124) plane (light-green

color), this tiny expansion of [124] distances could be correlated with the extent of thermal flipping of the OPTz moiety. This is the same as the other peak shifts in the PXRD patterns. Notably, the “flipping” structure showed a remarkable enlargement of pore aperture for the diffusion of CO₂ and C₂H₂ compared to the “static” structure, which indicated that the “flipping” structure allowed accelerating the diffusion of CO₂ and C₂H₂ in response to increasing temperature.

Reviewer-only Figure 1. Structure related to (124) facet in the unit cell of **FDC-3a**.

6. In this paper, the authors focus on the adsorption or diffusion of guest molecules in temperature-responsive FDC-3, and only illustrate by variable-temperature XRD and solid-state NMR that the ligands in the temperature-elevated FDC-3a are flipped, which affects the adsorption of guest molecules on FDC-3a. It is suggested to provide pure-component static and kinetic adsorption experiments of the material at different temperatures to further demonstrate the experimental phenomenon.

We appreciate the reviewer for this constructive comment. Indeed, we have performed the pure-component static adsorption measurements, namely, the adsorption isobars, in the initial manuscript. When we measured the isobar experiments, the exposure time of every temperature was not manually regulated, this meant that all the data in the figure were in their near-equilibrium states. This is why the adsorption isobars can be recognized as static adsorption at different temperatures.

According to the suggestion from the reviewer, we have performed the pure-component kinetic adsorption experiments at different temperatures. We set the relative pressure to be $P/P_0 = 1.0$ and the exposure time of every temperature to be 100 s. By this parameter, we obtained the kinetic adsorption curves of CO_2 and C_2H_2 at different temperatures. The curves exhibited three characteristic features: (1) The adsorption amounts for both CO_2 and C_2H_2 were lower than the amounts in their corresponding isobar curves. (2) The T_{max} for both CO_2 and C_2H_2 slightly shifted to higher temperatures. The above two features indicated that the kinetic factors were key to affecting the adsorption behaviors of CO_2 and C_2H_2 . (3) The selectivity switch was also observed in the kinetic adsorption, which further proved that the synergistic effect of diffusion regulation and host-guest interaction caused the temperature-switchable selectivity even in the kinetic conditions.

We have added the kinetic adsorption result to the revised Supplementary Materials as Fig. S15. We also added several sentences to the revised manuscript to discuss this result:

*“Although the above-mentioned sorption curves already revealed an apparent difference in the adsorption amounts of CO_2 and C_2H_2 , they were not able to reflect the differences in the adsorption kinetics. Therefore, we performed kinetic adsorption of CO_2 and C_2H_2 at different temperatures by **FDC-3a** (Supplementary Fig. S15). The adsorption amounts for both CO_2 and C_2H_2 were lower than the amounts in their corresponding isobar curves, whereas the T_{max} for both CO_2 and C_2H_2 slightly shifted to higher temperatures. These results indicated that the kinetic factors were key to affecting the adsorption behaviors of CO_2 and C_2H_2 . On the other hand, the switching of selectivity was also observed in the kinetic adsorption, which further proved that the cooperativity of diffusion regulation and host-guest interaction caused the temperature-switchable selectivity even in the kinetic conditions.”.*

Supplementary Figure S15. Kinetic adsorptions of CO₂ and C₂H₂ at different temperatures. The relative pressure is set to be $P/P_0 = 1.0$ and the exposure time of every temperature is fixed to be 100 s.

7. The author described in the manuscript that the pore size of the material is 2.9 Å, however, it can adsorb CO₂ (3.3 Å) and C₂H₂ (3.3 Å) at relatively high temperatures, indicating a change in the pore size of the material. Based on this, the single component adsorption isotherms at lower temperatures and the pore size of material at different temperatures should also be presented in the manuscript.

In the initial submission, we have already included the single-component adsorption isotherms of CO₂ and C₂H₂ in the Supplementary Materials as Fig. S12. The temperature range was 200 to 360 K (20 K interval). Because the sublimation point of CO₂ and the boiling point of C₂H₂ were 194.5 and 188.4 K, respectively, we determined the lowest temperature for measuring the adsorption isotherms to be 200 K, a slightly higher temperature than the sublimation/boiling point of the gas to prevent the gas condense at the surface of the PCP. Indeed, because of the low adsorption rate, CO₂ and C₂H₂ easily condensed at low temperatures (195 K for CO₂ or 189 K for C₂H₂) and high relative pressure (P/P_0 around 1.0). Therefore, the lowest temperature for measuring the single-component adsorption isotherms was 200 K as we provided in Supplementary Fig. S12.

Because the pore size was always dynamic due to the thermal flipping of the OPTz moiety at different temperatures, we were not able to provide an “accurate” pore-size value. However, we could deduce the pore size from the adsorption behavior of the PCP. As shown in Supplementary Fig. S11, our PCP started to adsorb CO₂ or C₂H₂ from 200 K, which meant that the pore size was slightly higher than 3.3 Å (the kinetic diameter of CO₂ or C₂H₂) at 200 K. On the other hand, our PCP did not adsorb N₂, CO, CO₂, C₂H₂, O₂, Ar, C₂H₄, and C₂H₆ from 200 to 360 K (the slight adsorption of N₂ at 80 to 150 K was probably due to the surface condensation), indicating that the pore size was smaller than 3.47 Å (the kinetic diameter of O₂, the smallest kinetic diameter among these gases) in this temperature range. Therefore, the pore sizes at 200 to 360 K ranged from 3.3 to 3.47 Å.

8. When calculating the IAST selectivity, the corresponding fitting model, fitting parameters, and correlation coefficients are missing.

According to the suggestion from the reviewer, we have added the calculation details of the IAST selectivity to the revised Supplementary Materials in the methods section and Supplementary Tables S4 and S5.

“Calculation of the IAST selectivity

To calculate the IAST selectivity, the single-component CO₂ and C₂H₂ adsorption isotherms for the adsorbate at different temperatures were first fitted to the dual-site Langmuir-Freundlich equation (equation 1):

$$q = n_{m1} \frac{b_1 P^{(1/t_1)}}{1 + b_1 P^{(1/t_1)}} + n_{m2} \frac{b_2 P^{(1/t_2)}}{1 + b_2 P^{(1/t_2)}} \quad (1)$$

where q is the amount adsorbed per mass of material (mmol g⁻¹); P is the total pressure (kPa) of the bulk gas at equilibrium with the adsorbed phase; n_{m1} and n_{m2} are the saturation uptakes (mmol g⁻¹) for sites 1 and 2, respectively; b_1 and b_2 are the affinity coefficients (kPa⁻¹) for sites 1 and 2, respectively; t_1 and t_2 represent the deviations from the ideal homogeneous surface (dimensionless) for sites 1 and 2, respectively.

Then, the IAST selectivity (S) is defined as:

$$S = \frac{q_1 / q_2}{p_1 / p_2} \quad (2)$$

where q_1 and q_2 are the molar loadings in the adsorbed phase in equilibrium with the bulk gas phase with partial pressures p_1 and p_2 (Supplementary Tables S4 and S5).”.

Section 3: Calculation of IAST selectivity

Supplementary Table S4. The dual-site Langmuir-Freundlich fitting parameters of CO₂ sorption data at different temperatures.

	200 K	220 K	240 K	260 K	280 K	300 K	320 K	340 K	360 K
n_{m1}	0.31095	0.25254	0.54276	0.29568	0.29568	1.12368	4.7692	0.36883	0.29707
n_{m2}	2.39602	5.90028	3.17592	1.66608	1.66608	1.12368	4.79664	0.36883	0.29707
b_1	1.21512	0.98529	1.99E-6	1.864E-7	1.86E-7	0.00931	6.62E-4	0.01091	0.00661
b_2	0.00226	0.0039	0.02641	0.02222	0.02222	0.00931	6.62E-4	0.01091	0.00661
t_1	1.15931	1.04777	0.31455	0.26451	0.26451	1.0942	0.80539	0.93055	0.96219
t_2	0.81998	0.99113	1.33953	1.10487	1.10487	1.0942	0.80539	0.93055	0.96219
R^2	0.9996	0.9996	0.9998	0.9999	0.9998	0.9998	0.9983	0.9980	0.9979

Supplementary Table S5. The dual-site Langmuir-Freundlich fitting parameters of C₂H₂ sorption data at different temperatures.

	200 K	220 K	240 K	260 K	280 K	300 K	320 K	340 K	360 K
n_{m1}	12.9366 5	0.45545	0.32295	35.5084 2	52.8838 1	58.2270 8	4.7692	2.94161	1.36217
n_{m2}	0.25777	0.3187	16.5578 1	0.31837	0.3599	0.13565	4.79664	2.94161	1.36217
b_1	0.00442	2.18E-4	2.39E-4	0.00121	6.03E-4	3.45E-4	6.62E-4	0.00182	0.00365
b_2	2.33762 E-5	0.18689	0.00403	7.54E-6	1.61E-6	1.73E- 12	6.62E-4	0.00182	0.00365
t_1	2.44582	0.49301	0.49949	1.70374	1.4386	1.1384	0.80539	0.9224	0.95457
t_2	0.42166	1.72121	2.29703	0.36048	0.31937	0.15287	0.80539	0.9224	0.95457
R^2	0.9989	0.9997	0.9994	0.9997	0.9995	0.9996	0.9983	0.9994	0.9991

9. According to the previous article (Nature Communications, 2023, 14, 3789), is it possible that the guest molecules form tetramers in the pores when the material adsorbs guest molecules at low temperatures? What is the packing density? Is the force dominated by the guest-guest or host-guest interactions?

We appreciate the reviewer for this comment. Accordingly, we have checked the *Nature Communication* paper by D. Zhao *et al.* and found that their MOF possessed a large pore that allowed four gas molecules to form tetramers to facilitate the thermodynamic separation of CO₂ and C₂H₂. By contrast, our PCP was essentially different from their case because the pore aperture of our PCP was substantially smaller than theirs, whereas one “cage” in our PCP could only contain one CO₂ or C₂H₂ molecule. Therefore, no aggregated structures of gas molecules could be formed in our PCP, thus ensuring that the diffusion of CO₂ or C₂H₂ was strictly

regulated to show the observed selectivity at different temperatures. Of course, we recognized that the *Nature Communication* paper performed nice research on selective adsorption of CO₂/C₂H₂. Therefore, we have cited this paper as Ref. 17 in the revised manuscript.

The packing density was calculated to be 1.56 and 0.54 g cm⁻³ for CO₂ (240 K) and C₂H₂ (320 K), respectively, according to the ratio of the adsorbed gas mass to the pore volume of PCP. As the gas molecules in our PCP were monomers, the driving force was determined by the host-guest interactions.

=====

10. For the theoretical calculations, how to confirm the reliability of the Universal force field in MC simulations? The details for calculating the diffusion pathways should be provided. For simulating diffusion, it is reasonable to use the supercell model instead of a unit cell, since the pore aperture plays a significant effect.

=====

We appreciate the reviewer for this comment. MC simulations were used only to find the initial adsorption positions of gas molecules in the PCP. Because there are no open metal sites in our PCP, the universal force field (UFF) is useful to describe the van der Waals interactions between gas molecules and the PCP framework. These initial adsorption positions obtained by MC simulations with UFF will be further refined by calculations using the density functional theory method. For the calculation of the diffusion barrier, we evaluated the single-point energy of each loading position of the obtained guests-loading saturated **FDC-3a**, as described on Page 6 of the Supplementary Materials. Because the use of a supercell could be time-consuming and a unit cell is sufficiently large to consider the influence of neighboring ligands on gas diffusion, we considered using only a unit cell despite that a larger supercell could be more reasonable.

According to the suggestion from the reviewer, we have performed additional experiments and revised our manuscript critically. We believe the revision has clarified all the issues raised by the reviewer. We sincerely hope the reviewer can understand our positive situation regarding the revision and be satisfied with the additional experiments and revisions we performed. We appreciate the reviewer for the above constructive comments that improve the manuscript.

=====

REVIEWER COMMENTS

Reviewer #1 (Remarks to the Author):

The authors have revised the manuscript and I have no longer concerns with respect to the crystallographic data. With respect to the proposed flip-flop motion, I am still not convinced. According to the authors' calculations, the gate-blocking moiety only rotates back and forth by about 20-30°. This is not the situation described as flip-flop motion, it is rather a wagging mode of the OPTz unit. A full rotation seems very unlikely as the required energy increases significantly at larger rotation angles (see Supplemental Fig. S1).

I still think the work is worth to be published, but the movement of the OPTz needs to be discussed in some more detail. Interestingly, the authors propose a flip-flop motion but do not discuss that motion in the main manuscript, it is only very shortly addressed in the SI without further discussion. Once that point is addressed satisfactorily, the manuscript should be acceptable for publication.

Reviewer #2 (Remarks to the Author):

Authors revised the manuscript considering comments by Reviewer #2.

However, I recommend the further revision for better paper.

Also, you need to search temperature dependence of breathing effect and gate effect on MOF.

1: MIN-2 size

C. E. Webster et al, JACS, 1998, 120, 5509

K. S. W. Sing and R. T. Williams, Part. Part. Syst. Charact. 2004, 21, 71

2: Discussion on selectivity difference between component gas experiment and mixed gas one must be done. The discrepancy is quite large and then authors may discuss the reason.

CO₂-C₂H₂ interaction must be taken into account.

Response to Referee #1

The authors have revised the manuscript and I have no longer concerns with respect to the crystallographic data. With respect to the proposed flip-flop motion, I am still not convinced.

According to the authors' calculations, the gate-blocking moiety only rotates back and forth by about 20-30°. This is not the situation described as flip-flop motion, it is rather a wagging mode of the OPTz unit. A full rotation seems very unlikely as the required energy increases significantly at larger rotation angles (see Supplemental Fig. S1).

I still think the work is worth to be published, but the movement of the OPTz needs to be discussed in some more detail. Interestingly, the authors propose a flip-flop motion but do not discuss that motion in the main manuscript, it is only very shortly addressed in the SI without further discussion.

Once that point is addressed satisfactorily, the manuscript should be acceptable for publication.

We appreciate the comments from the reviewer endorsing the revision. Regarding the concern raised by the reviewer, we have taken significant consideration of the comments and made a full explanation. We believe that the explanation has clarified all the comments from the reviewer.

We agree with the reviewer that “wagging” is better to be used herein. Indeed, the OPTz moiety does not fully rotate. We used the term “flipping” here because we hope to keep consistent with our previous work, where flipping mode exists (*Science* **2019**, 363, 387).

We are also endeavoring to develop more methods to characterize the flip-flop motion because we know this is crucial for future mechanism studies. However, this was much more difficult than we expected. Before this work, we were only able to use the calculation and VT-PXRD to indicate the occurrence of this motion because the transient and dynamic behavior is considerably difficult to characterize. This is why we did not discuss the flipping motion more in our previous papers (*Science* **2019**, 363, 387; *Nature* **2022**, 611, 289) because we would like to keep the preciseness in the papers. Fortunately, we got in touch with NMR experts and we used NMR to characterize the flipping motion in this work. However, NMR is still not direct evidence, therefore, we tended not to discuss more about the flipping motion in the current version. However, with the reviewer’s suggestion in mind, we have started to collaborate with other experts to characterize the flipping motion. For instance, we are trying the XFEL

technique in Japan because it can reveal a transient behavior with \sim ps resolution. We would like to report the results in future manuscripts.

We sincerely hope the reviewer can understand our positive situation regarding the revision and be satisfied with the explanation we performed. We appreciate the reviewer for the above constructive comments that improve the manuscript.

Response to Referee #2

Authors revised the manuscript considering comments by Reviewer #2.

However, I recommend the further revision for better paper.

Also, you need to search temperature dependence of breezing effect and gate effect on MOF.

We appreciate the comments from the reviewer endorsing our work. Regarding the concern raised by the reviewer, we have taken significant consideration of all the comments and suggestions from the reviewer and revised our manuscript accordingly. We believe that the revised manuscript has clarified all the requests and comments from the reviewer.

1: MIN-2 size

C. E. Webster et al, JACS, 1998, 120, 5509

K. S. W. Sing and R. T. Williams, Part. Part. Syst. Charact. 2004, 21, 71

We appreciate the reviewer for providing the literature. Accordingly, we have added the MIN-2 parameters to Table S3 in the revised Supplementary Materials. We also added the two papers as Ref. S23 and S33.

2: Discussion on selectivity difference between component gas experiment and mixed gas one must be done. The discrepancy is quite large and then authors may discuss the reason.

CO₂-C₂H₂ interaction must be taken into account.

We appreciate the reviewer for this constructive comment. At low temperatures, the diffusion of both CO₂ and C₂H₂ was regulated, and the gas separation was governed by the diffusion-rate difference. CO₂ showed a faster diffusion rate than C₂H₂, resulting in high CO₂/C₂H₂ selectivities under non-equilibrium states. On the other hand, at high temperatures, the cooperativity of CO₂-C₂H₂ interaction and gas-framework interaction amplified the selective adsorption of C₂H₂, rendering a higher C₂H₂/CO₂ selectivity than the expected one from the single-gas adsorption.

According to the suggestion from the reviewer, we have added several sentences in the revised manuscript to discuss the selectivity differences:

“Notably, the selectivities of the mixed-gas separation were higher than the ones predicted from the single-gas adsorption. At low temperatures, the diffusion of both CO₂ and C₂H₂ was regulated, and the gas separation was governed by the diffusion-rate difference. CO₂ showed a faster diffusion rate than C₂H₂, resulting in high CO₂/C₂H₂ selectivities under non-equilibrium states. On the other hand, at high temperatures, the cooperativity of CO₂-C₂H₂ interaction and gas-framework interaction amplified the selective adsorption of C₂H₂, rendering a higher C₂H₂/CO₂ selectivity than the expected one from the single-gas adsorption.”.

According to the suggestion from the reviewer, we have revised our manuscript critically. We believe the revision has clarified all the issues raised by the reviewer. We sincerely hope the reviewer can understand our positive situation regarding the revision and be satisfied with the revisions we performed. We appreciate the reviewer for the above constructive comments that improve the manuscript.

REVIEWERS' COMMENTS:

Reviewer #2 (Remarks to the Author):

I recommend publication, because the authors improved their manuscript considering my comments.